# Circulating Inflammatory Cytokine Associated with Poor Prognosis in Moyamoya Disease: A Prospective Cohort Study

**DOI:** 10.3390/jcm12030823

**Published:** 2023-01-19

**Authors:** Wei Liu, Jian Sun, Zhiyong Shi, Zheng Huang, Lebao Yu, Haibin Du, Peicong Ge, Dong Zhang

**Affiliations:** 1Department of Neurosurgery, Beijing Tiantan Hospital, Capital Medical University, Beijing 100070, China; 2Department of Neurosurgery, Beijing Changping District Hospital, Beijing 100071, China

**Keywords:** moyamoya disease, TNF-α, IL-6, inflammatory cytokines, RNF213

## Abstract

Inflammation is a key factor in the development of moyamoya disease. However, the cytokine distribution in moyamoya disease and its impact on prognosis remain unclear. A total of 204 patients with moyamoya disease were enrolled in this study. The peripheral blood was analyzed for baseline data and cytokines, which included IL-6, IL-1β, IL-2R, IL-8, and TNF-α. Patients with the RNF213 mutation and those without the mutation were compared in terms of their differences in cytokines. A mRS score ≥ 2 was defined as a poor prognosis, and a mRS score < 2 was described as a good prognosis, and differences in cytokines were compared between the two groups. Regression analysis was performed to identify markers affecting prognosis. TNF-α and IL-6 levels were higher in the group without the RNF213 mutation compared to the mutation group. Multivariate stepwise regression analysis indicated that the G3 subgroup of IL-6 and the G4 subgroup of TNF-α were the independent risk factors for adverse prognosis in adults with moyamoya disease (OR 3.678, 95% CI [1.491, 9.074], *p* = 0.005; OR 2.996, 95% CI [1.180, 7.610], *p* = 0.021). IL-6 and TNF-α were associated with poor prognosis in adult patients with moyamoya disease.

## 1. Introduction

Moyamoya disease (MMD) is a rare progressive cerebrovascular disease that is mainly characterized by progressive stenosis and occlusion at the ends of the bilateral internal carotid arteries and Willis rings [1,2], resulting in the creation of abnormal collateral vessels at the base of the brain, with the compensatory abnormal vascular network, the so-called ‘moyamoya vessels’ [3]. MMD was first defined by Suzuki and Takahisa in Japan in 1969, with a high prevalence rate in East Asian populations and an incidence rate of 0.94/100,000 person-years in the Japanese population [4]. According to recent studies, the incidence of MMD in mainland Chinese populations is 1.14 per 100,000 person-years [5]. This value is lower in Europe and the United States, where the incidence of moyamoya disease is 0.07/100,000 person-years in Europe and 0.293/100,000 person-years in the United States [6,7]. However, the incidence of moyamoya disease in various places shows an increasing trend year by year.

Various physiological and biochemical processes are regulated by inflammation. Previous studies have shown that inflammation is closely related to the incidence and progression of MMD [8,9,10]. Several cytokines are elevated in MMD compared to normal subjects, including IL-17, TNF-α, IL-6, and IL-23 [11]. However, the prognostic impact of cytokines in MMD remains unclear. The routine measurement of inflammatory markers to predict functional outcomes in MMD is not yet recommended by updated guidelines, and there are limited data available on the prognostic contribution of inflammatory co-markers in MMD.

Ring finger protein 213 (RNF213), an E3 ubiquitin ligase, is reported to be a susceptibility gene for MMD. The p.R4810 k mutation in RNF213 is the prevalent variant in East Asian MMD patients [12,13]. Previous studies have shown that the transcription of RNF213 is regulated by cytokines [14,15], yet the linkage between polymorphisms in RNF213 and cytokines remains unclear in clinical practice.

We collected blood samples from patients with MMD from Beijing Tiantan Hospital, Capital Medical University, to identify inflammatory markers that can predict the functional prognosis of MMD and facilitate early identification and assessment of high-risk patients. In addition to establishing the role of cytokines, the RNF213 mutation was also studied, and its relation to the distribution of cytokines was evaluated in MMD.

## 2. Methods

### 2.1. Study Design and Subjects

We consecutively collected 259 patients with MMD who visited Beijing Tiantan Hospital, all of whom had undergone digital subtraction angiography (DSA) and whose diagnosis followed the updated guidelines for managing MMD. We excluded 48 pediatric patients, 7 patients aged >60 years. Finally, 204 patients with MMD were enrolled in the current study. The inclusion and exclusion criteria followed are shown in Appendix A. The laboratory testing department at Beijing Tiantan Hospital verified that all the cytokine data were of clinical diagnostic significance. The primary outcome was a poor functional outcome at post-operative discharge, defined as a mRS score ≥ 2 [16]. This study was approved by the Ethics Committee of Beijing Tiantan Hospital. A documented informed consent agreement has been signed by all patients.

### 2.2. Clinical Data Collection

We collected risk factors that may be associated with prognosis in MMD, including demographic data, medical history, clinical features, modified Rankin Scale (mRS) score, imaging findings, and laboratory tests. Demographic information included age and gender. Hypertension (self-declared history of hypertension, or use of any BP-lowering medication, with a systolic blood pressure ≥ 140 mmHg or diastolic blood pressure ≥ 90 mmHg), diabetes (self-declared history of diabetes, or use of any glucose-lowering medication, with any fasting blood glucose level ≥ 7.0 mmol/L) and hyperlipidemia (self-declared history of hyperlipidemia, or use of any lipid-lowering medication, with LDL cholesterol ≥ 3.37 mmol/L, HDL cholesterol < 1.04 mmol/L, triglycerides ≥ 1.7 mmol/L, or total cholesterol ≥ 5.17 mmol/L) were considered as medical history. Clinical features were summarized as symptom manifestations, surgical approach, and body mass index (BMI). The Suzuki stage was judged by DSA on the surgical side. The side with the highest bilateral Suzuki stage was taken to represent the patient’s total Suzuki stage. According to the Suzuki stage, stages 0–3 are defined as the early stages, and stages 4–6 are defined as the late stages. Laboratory test data included white blood cell count (WBC), red blood cell count (RBC), hemoglobin concentration (HGB), platelet count (PLT), urea, uric acid (UA), triglyceride (TG), total cholesterol (CHO), and homocysteine (Hcy).

Blood samples were collected from the 166 patients mentioned above after written informed consent was obtained. Genomic DNA was extracted from 166 patients using the QIA amp blood kit (QIAGEN, Hilden, Germany). Primers were designed as follows: RNF213–4810F(rs112735431):5′-GCCCCTCTAGCACAC-3′; and RNF213–4810R: 5′-AGCTGTGGCGAAAGCTTCTA-3′. In total, 36 patients with the RNF213 rare variant p.R4810 k (rs112735431) were found in the 166 sequenced patients.

### 2.3. Statistical Analysis

IL-6 and TNF-α were divided into two groups, normal and high, according to their in-human normal value criteria (3.40 and 8.10 pg/mL, respectively). These criteria were formulated by the Clinical Laboratory Department of Beijing Tiantan Hospital with clinical diagnostic significance. Within the normal and high groups, the groups were further divided into two groups according to the median. IL-6: group 1 (G1): ≤2.00 pg/mL; group 2 (G2): 2.00–3.40 pg/mL; group 3 (G3): 3.40–4.83 pg/mL; group 4 (G4): >4.83 pg/mL. TNF-α: group 1 (G1): ≤6.24 pg/mL; group 2 (G2): 6.24–8.10 pg/mL; group 3 (G3): 8.10–9.26 pg/mL; group 4 (G4): >9.26 pg/mL. IL-1β was divided into a normal group (0–5.00 pg/mL) and a high group (>5.00 pg/mL) according to the level of normal values.

Statistical analysis was performed using SPSS (version 26.0, IBM, Armonk, NY, USA). To analyze the relationship between the p.R4810K mutation in RNF213 with cytokines, the raw baseline was assessed using *t*-tests for continuous variables and chi-square tests for categorical variables. To evaluate the relationship between cytokines and prognosis, we performed univariate and multifactorial regression analysis on cytokines and other confounding factors. We performed three regression models to assess risk factors for poor outcomes in moyamoya disease patients: the crude model was the unadjusted model. Model 1 adjusted for sex and age. Model 2 further adjusted for hypertension, hyperlipidemia, diabetes, Image stage, white blood cell count, hemoglobin, triglycerides, total cholesterol, Hcy, BMI, and surgical approach [7]. Unadjusted and adjusted risk ratios (OR) and their 95% confidence intervals (CI) were calculated. Receiver operating characteristic (ROC) curve analysis was performed to evaluate the utility of a continuous biomarker. The choice of cut-off point in the ROC curve analysis is based on the maximum value of the Uden index.

## 3. Results

### 3.1. Baseline Characteristics

Baseline characteristics of subjects grouped according to prognosis are shown in Table 1. A cohort of 204 patients with MMD was enrolled: 81 men and 123 women, with a median age of 44 years. There were 101 patients in the group with a mRS score < 2 and 103 patients with a mRS score ≥ 2. Compared to those in the group with mRS scores < 2, patients in the group with mRS scores ≥ 2 were older (*p* < 0.001), more patients presented with bilateral MMD (92.233% vs. 83.168%, *p* < 0.05), and a higher proportion had hypertension (43.689% vs. 22.772%, *p* < 0.05). IL-6 levels differed between the two groups (*p* < 0.05). A comparative analysis was performed, and a statistical difference between IL-6 and TNF-α among diabetics and non-diabetics was found. The results showed that the diabetic group had higher IL-6 levels than the non-diabetic group (5.785 ± 6.837 vs. 2.904 ± 2.650 pg/mL, *p* < 0.05). There were no significant differences observed in other clinical features, including sex, image stage, history of diabetes mellitus, hyperlipidemia, clinical manifestation, body mass index, IL-1β, IL-2R, IL-8, and other laboratory results. The differences between the above-mentioned inflammatory factors and between the diabetic and non-diabetic groups are shown in Figure 1 and Appendix A.

### 3.2. Higher TNF-α and IL-6 Levels in the p.R4810K Variant of RNF213

The enrolled patients were divided into mutation and no mutation groups based on the presence of the p.R4810 k mutation in RNF213. Analysis of TNF-α and IL-6 as continuous variables showed higher levels of IL-6 in the no mutation group compared to the mutation group (3.437 ± 3.098 vs. 2.496 ± 1.460, *p* < 0.05). When investigating TNF-α and IL-6 in groups, statistical analysis showed that a higher proportion of the patients in the no-mutation group had high TNF-α values compared to the mutation group (26.05% vs. 10.64%, *p* < 0.05). Other cytokine levels were not significantly different between the two groups. The differences in IL-6 and TNF-α in the mutant and non-mutant groups are shown in Figure 2 and Appendix A.

### 3.3. IL-6 and TNF-α Are Associated with Poor Functional Prognosis after Surgery for Adult MMD

Inflammatory factors were analyzed in univariate regression along with other factors that may affect the prognosis of MMD. Univariate logistic regression analysis revealed that IL-6 and TNF-α subgroups were associated with an increased risk of poor functional outcome after surgery in adults with MMD. In the G3 subgroup of IL-6 and the G4 subgroup of TNF-α, the risk of poor prognosis of MMD was higher than that of the G1 subgroup. (OR 3.678, 95% CI [1.491, 9.074], *p* = 0.005; OR 2.996, 95% CI [1.180, 7.610], *p* = 0.021). Univariate logistic regression analysis showed that advanced age (OR 1.081, 95%CI [1.047, 1.116], *p* < 0.001) and hypertension (OR 2.631, 95%CI [1.435, 4.826], *p* < 0.05) Risk factors for poor functional outcome after MMD surgery. The specific univariate logistic regression analysis results are shown in Appendix A.

Trend regression analysis showed that with the increase in TNF-α level in blood, the risk of poor prognosis in MMD patients gradually increased, with a significant trend (P for trend = 0.013). After adjusting more confounding factors by multivariate logistic regression analysis, the results showed that the G4 subgroup of TNF-α and the G3 subgroup of IL-6 were independent risk factors for poor functional prognosis in MMD (OR 4.626, 95% CI [1.493, 14.337], *p* = 0.008; OR 4.794, 95% CI 4.794, *p* = 0.004). The results of the stepwise regression analysis are shown in Table 2. The ROC curve showed that compared with the Crude model and model 1, the predictive value of model 2 for the poor prognosis of MMD was always significantly improved. (AUCs of TNF-α were 0.609, 0.724, 0.743; AUCs of IL-6 were 0.569, 0.713, 0.735). The risk forest plot and stepwise regression ROC curve of TNF-α and IL-6 levels and MMD prognosis are shown in Figure 3. By separately incorporating IL-6 and TNF-α, ROC curve analysis showed that TNF-α levels; 6.620 ng/mL could predict adverse functional outcomes in adults with MMD, with a sensitivity of 64.1%, a specificity of 56.4%, and an area under the curve is 0.609. The combination of TNF-α and IL-6 did not increase the predictive power of the model. The results of the ROC analysis are shown in Figure 4.

The crude model was the unadjusted model. Model 1 adjusted for sex and age. Model 2 further adjusted for hypertension, hyperlipidemia, diabetes, Image stage, white blood cell count, hemoglobin, triglycerides, total cholesterol, Hcy, BMI, and surgical approach.

## 4. Discussion

Our cohort study analyzed multiple cytokines to demonstrate the comprehensive role of cytokines in MMD. Rather than exploring the relationship between inflammation and the pathogenesis of MMD, as in previous studies, the focus of the present study was on the impact of inflammation on the prognosis of MMD. Our study shows that the severity of systemic inflammation affects functional outcomes in MMD. The prognostic marker of inflammation identified in the current study was TNF-α, which at pre-operative blood levels of >9.24 pg/mL, was significantly associated with an increased risk of adverse functional outcomes after surgery and was an independent risk factor for poor prognosis in MMD. Our study demonstrates the relationship between RNF213 polymorphism and cytokine levels in MMD patients. TNF-α and IL-6 in patients without p.R4810K mutation of RNF213 were higher than those in the mutation group. The cytokine IL-6 acts on a variety of target cells, activates intracellular tyrosine kinases, and has been implicated in the activation of many pathways, including the signal transducer and activator of transcription (STAT) pathway, the MAPK pathway, the PI3K pathway, and the insulin receptor substrate (IRS) [17]. PI3K-Akt and PKR are two major upstream pathways that regulate RNF213 expression in endothelial cells and are associated with endothelial cell proliferation and angiogenic potential [14,18]. Cytokine stimulation significantly induces RNF213 transcription through the PI3K-Akt and PKR pathways. Thus, IL-6 may regulate angiogenesis in MMD patients through the regulation of RNF213 transcription, consequently affecting their prognosis.

Hypercoagulation is an essential manifestation of inflammation, and cytokines play a decisive role in this process [19]. Increased levels of IL-6 in peripheral blood and tissues can stimulate the production of platelet-derived growth factors, leading to a thrombosis-promoting state [20]. Thrombosis is tightly associated with the development of intracranial arterial thickening. In autopsy specimens of internal carotid arteries from patients with MMD, a thrombus was present in 16 out of 22 patients [21]. In addition to the thickened intima, occlusion of blood vessels is a vital component of MMD. The susceptibility to thrombotic disease is therefore elevated in patients with MMD [22].

According to studies by Su et al., ischemia and hypoxia, oxidative stress, vascular occlusion, and inflammation induce dysregulation of IL-6, leading to the hepatic generation of acute phase proteins that facilitate leukocyte recruitment and thrombosis, eventually causing various cardiovascular diseases, including ischemic stroke [23]. Enhanced levels of blood IL-6 are associated with an increased risk of stroke and contribute to racial differences in stroke through the influence of inflammatory risk factors [24]. Elevated plasma IL-6 is reported as a feature of post-stroke delirium [25]. Furthermore, in the case of large vessel occlusion in patients with acute ischemic stroke, high levels of IL-6 within 24 h of thrombus recanalization are associated with ineffective reperfusion [26]. It has been shown that complete or near-complete reperfusion following thrombus recanalization is associated with lower admission levels of IL-6, suggesting that IL-6 may predict favorable outcomes in patients with acute ischemic stroke [27]. These findings suggest that IL-6 may be a predictor of prognosis in patients with ischemic stroke. Thus, in MMD, IL-6 may induce an exacerbation of the pre-operative ischemic state by promoting thrombosis as well as amplifying inflammation and impeding revascularization following bypass surgery, leading to a poorer functional outcome. Previous studies have reported that IL-6 can promote glucagon secretion and induce insulin resistance. Elevated circulating levels of IL-6 are an independent predictor of type 2 diabetes development [28,29]. Since elevated levels of IL-6 have been found in patients with co-morbid diabetes mellitus, it may not be specific to MMD. However, diabetes is also a cardiovascular risk factor, and its impact on the prognosis of MMD has been reported in previous studies [30], and this might be seen as a way in which IL-6 could influence the functional outcome of MMD.

TNF-α is an inflammatory cytokine that is expressed at low levels in the healthy central nervous system. However, previous studies have shown that following experimental and human ischemic stroke, the expression levels of TNF-α and its two identified receptors (TNFR1 and TNFR2) [31] are elevated and can influence the progression of infarct [32,33,34]. Common polymorphisms in the TNF-α gene promoter result in elevated TNF-α levels, which appear to be associated with cardiovascular risk factors and ischemic stroke in Asians. Furthermore, plasma TNF-α levels were elevated in stroke patients compared to healthy controls [34]. Chen et al. also showed that TNF-α leads to endothelial cell apoptosis, resulting in blood–brain barrier disruption. Additionally, they suggested that activating NF-κB nuclear translocation induces increased TNF-α production, which may lead to increased barrier disruption [35]. One known piece of information is that the NF-κB pathway was found to be upregulated in MMD [36]. RNF213 induces NF-κB activation and mediates apoptosis, which is similar to TNF-α effects. Additionally, blood–brain barrier dysfunction has been found in patients with MMD, and TNF-α may exacerbate this outcome and thus affect the prognosis of MMD. The inclusion of mutations in RNF213 in stepwise regression analysis masked the predictive role of TNF-α for poor prognosis. This suggests that TNF-α acts synergistically with RNF213, possibly through the same mechanism. This is in line with earlier studies showing that cytokines regulate the transcription of RNF213. TNF-α may also influence MMD prognosis by regulating the transcription of RNF213.

In a Japanese epidemiological survey, patients with RNF213 mutations accounted for only 73%, and this proportion was even lower in China [37]. Therefore, factors other than gene mutations such as inflammation may be involved in the pathogenesis of MMD. MMD cases with systemic sclerosis, autoimmune thyroiditis, and central nervous system infection have all been reported [38,39,40,41], and autoimmune abnormalities may be related to the pathogenesis of MMD [42]. In mutation-free MMD, inflammatory factors may be the main triggers of MMD, rather than genetic mutations, such as the pR4810K mutation in RNF213. Therefore, in the mutation-free group of MMD, the role played by inflammatory factors accounted for a larger proportion. This could explain why the levels of cytokines were higher in the no-mutation group compared to the mutant group. At the same time, this result further confirmed that the p.R4810K gene mutation of RNF213 is a susceptibility factor for MMD. This is due to the fact that RNF213 mutations can promote the development of MMD in patients with lower inflammatory factors. Our study showed that both RNF213 mutation and inflammatory factors are important in the pathogenesis of MMD, consistent with previous findings [10].

Our study also has some limitations. Firstly, the patient group was not compared with healthy controls; rather, the relationship between cytokines and prognostic-related indicators was compared between MMD subgroups. However, the expression patterns of the aforementioned inflammatory factors in MMD patients, and healthy controls have been compared in previous studies [11,43,44]. Secondly, no further analysis of the longer-term prognosis in relation to inflammatory factors has been undertaken, and further studies are required to collect data during follow-up. Thirdly, more research is needed to explore the exact role of inflammatory factors in the pathophysiology of MMD. Fourthly, the prognostic outcomes of this study did not take into account the reduced quality of life in patients with moyamoya disease due to transient ischemia, headache, and cognitive impairment, which may have a more serious impact on the individuals daily life, Therefore, more in-depth research is required. Finally, further validation in other cohorts is required because a single cohort from a single-center study has limited interpretability of the results.

## 5. Conclusions

TNF-α was associated with the risk of the poor short-term prognosis in adults with MMD. TNF-α and IL-6 levels were higher in the group without the pR4810K mutation of RNF213 compared to the mutation group. Our study demonstrates a comprehensive profile of cytokines in patients with MMD and identifies an inflammatory target for future research and treatment of MMD.

## Figures and Tables

**Figure 1 jcm-12-00823-f001:**
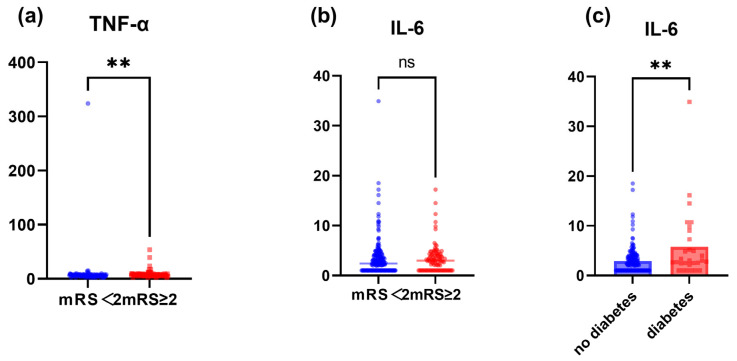
Comparison of serum TNF-α and IL-6 concentrations. (**a**) Differences in TNF-α between mRS < 2 and mRS ≥ 2 groups. (**b**) Differences in IL-6 between mRS < 2 and mRS ≥ 2 groups. (**c**) Differences in IL-6 between diabetes and no diabetes groups. ns, no significance; ** *p* < 0.01.

**Figure 2 jcm-12-00823-f002:**
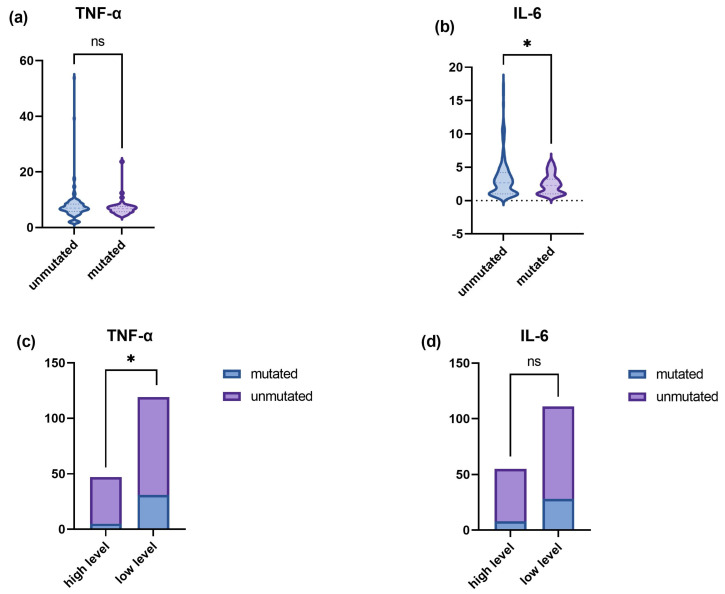
Comparison of IL-6 and TNF-α levels between p.R4810K mutated and unmutated in RNF213. (**a**,**b**) are the differences between the mutation group and the non-mutation group when TNF-α and IL-6 were analyzed as continuous variables. (**c**,**d**) are the differences between the mutation group and the non-mutation group when TNF-α and IL-6 are divided into a high-level group and a low-level group for analysis. ns, no significance; * *p* < 0.05.

**Figure 3 jcm-12-00823-f003:**
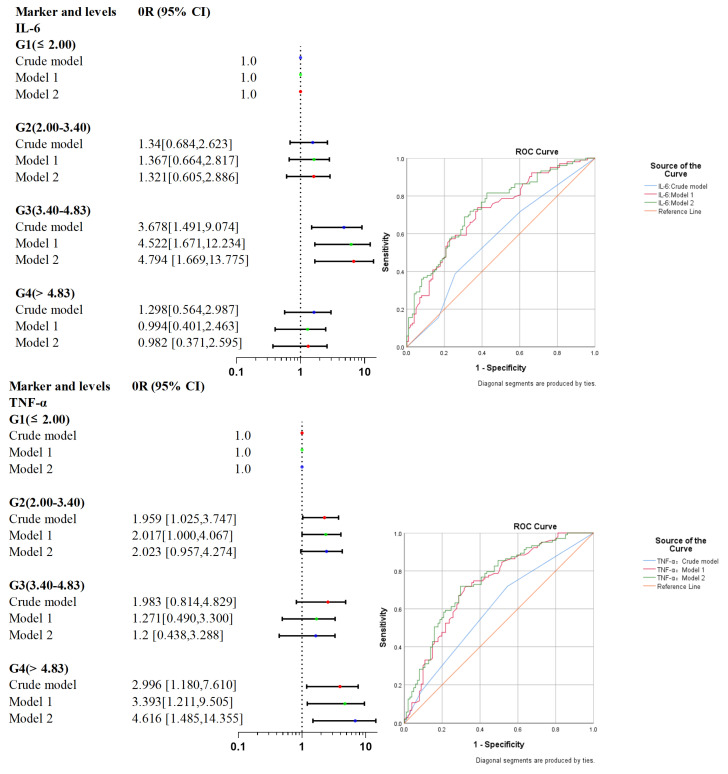
Risk in IL-6 and TNF-alpha grouping in correlation with the prognosis of MMD.

**Figure 4 jcm-12-00823-f004:**
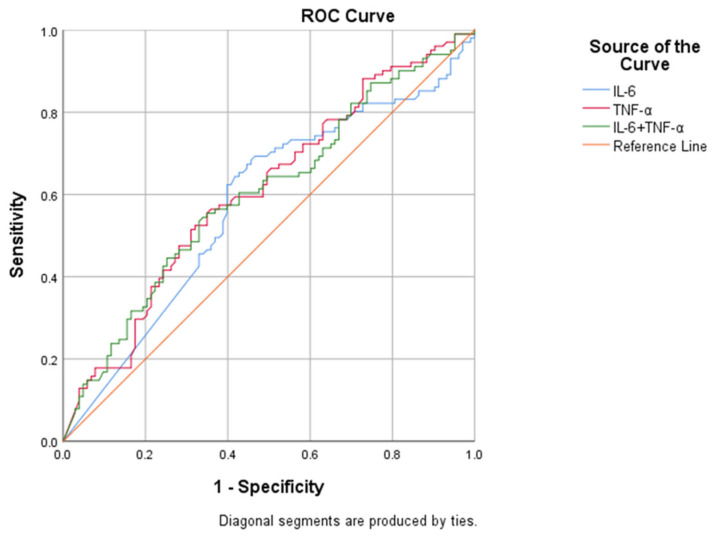
The risk forest plot of the relationship between IL-6, TNF-α and the prognosis of moyamoya disease and the ROC curve of the stepwise regression model.

**Table 1 jcm-12-00823-t001:** Baseline characteristics of 204 patients with moyamoya disease.

Variables	Value	ALL Patients (*n* = 204)	mRS < 2 (*n* = 101)	mRS ≥ 2 (*n* = 103)	*p*-Value
Age, median[IQR]	-	44.000 [34.000, 50.000]	38.000 [31.000, 47.000]	47.000 [40.000, 52.000]	<0.001
Gender/male, *n* (%)	-	81 (39.706)	37 (36.634)	44 (42.718)	0.375
Clinical manifestation, *n* (%)	-				
	ischemic	143 (70.098)	77 (76.238)	66 (64.078)	0.058
	hemorrhagic	61 (29.902)	24 (23.762)	37 (35.922)	
RNF213 mutation, *n* (%)					
	Unmutated	130/166 (78.313)	57/75 (76.000)	73/91 (80.220)	0.511
	mutated	36/166 (21.687)	18/75 (24.000)	18/91 (19.780)	
Unilateral or bilateral, *n* (%)					
	unilateral	25 (12.255)	17 (16.832)	8 (7.767)	0.048
	bilateral	179 (87.745)	84 (83.168)	95 (92.233)	
Image Stage, *n* (%)					
	early	127 (62.255)	65 (64.356)	62 (60.194)	0.54
	late	77 (37.745)	36 (35.644)	41 (39.806)	
suzuki stage, *n* (%)					
	1	1 (0.490)	1 (0.990)	0 (0.000)	<0.001
	2	56 (27.451)	32 (31.683)	24 (23.301)	
	3	70 (34.314)	32 (31.683)	38 (36.893)	
	4	32 (15.686)	13 (12.871)	19 (18.447)	
	5	29 (14.216)	14 (13.861)	15 (14.563)	
	6	16 (7.843)	9 (8.911)	7 (6.796)	
Hyperlipidemia, *n* (%)		30 (14.706)	12 (11.881)	18 (17.476)	0.259
Hypertension, *n* (%)		68 (33.333)	23 (22.772)	45 (43.689)	0.002
Diabetes, *n* (%)		29 (14.216)	11 (10.891)	18 (17.476)	0.178
BMI, median[IQR]		24.977 [22.491, 28.040]	25.352 [22.583, 28.040]	24.768 [22.266, 28.028]	0.549
WBC, median[IQR]		6.810 [5.620, 8.300]	6.940 [5.690, 8.510]	6.780 [5.520, 8.140]	0.456
RBC, mean (±SD)		4.605 ± 0.529	4.573 ± 0.574	4.637 ± 0.480	0.394
HGB, mean (±SD)		140.000 [129.000, 155.000]	137.000 [129.000, 154.000]	142.000 [130.000, 155.000]	0.639
PLT, median[IQR]		237.000 [204.000, 278.000]	245.000 [204.000, 279.000]	236.000 [204.000, 276.000]	0.827
PCT, median[IQR]		0.230 [0.200, 0.260]	0.230 [0.190, 0.260]	0.230 [0.200, 0.270]	0.876
Hcy, median[IQR]		10.560 [8.600, 13.100]	10.330 [8.160, 12.940]	10.810 [9.200, 13.400]	0.264
CHO, mean (±SD)		4.346 ± 0.938	4.398 ± 0.924	4.296 ± 0.949	0.438
TG, median[IQR]		1.220 [0.860, 1.590]	1.220 [0.820, 1.720]	1.220 [0.910, 1.530]	0.728
UA, median[IQR]		305.600 [253.200, 367.600]	301.200 [244.500, 363.400]	307.800 [274.000, 380.000]	0.064
Urea, median[IQR]		4.700 [3.800, 5.800]	4.700 [3.800, 5.800]	4.600 [3.800, 5.600]	0.801
Glu, median[IQR]		5.120 [4.710, 5.640]	5.060 [4.740, 5.490]	5.150 [4.710, 5.860]	0.175
IL-6, median[IQR]		2.430 [1.000, 3.990]	2.200 [1.000, 3.520]	2.980 [1.000, 4.190]	0.051
IL-6, *n* (%)					
	G1 (≤2.00)	69 (33.824)	40 (39.604)	29 (28.155)	0.035
	G2 (2.00–3.40)	69 (33.824)	35 (34.653)	34 (33.010)	
	G3 (3.40–4.83)	33 (16.176)	9 (8.911)	24 (23.301)	
	G4 (>4.83)	33 (16.176)	17 (16.832)	16 (15.534)	
TNF-α, median[IQR]		6.790 [5.460, 8.140]	6.350 [5.290, 7.650]	7.130 [6.060, 8.470]	0.007
TNF-α, *n* (%)					
	G1 (≤6.24)	75 (36.765)	46 (45.545)	29 (28.155)	0.058
	G2 (6.24–8.10)	76 (37.255)	34 (33.663)	42 (40.777)	
	G3 (8.10–9.26)	27 (13.235)	12 (11.881)	15 (14.563)	
	G4 (>9.26)	26 (12.745)	9 (8.911)	17 (16.505)	
IL-1β, *n* (%)					
	NORMAL (0–5.00)	162 (79.412)	81 (80.198)	81 (78.641)	0.783
	HIGH (>5.00)	42 (20.588)	20 (19.802)	22 (21.359)	
IL2R, mean (±SD)		332.564 ± 103.813	341.386 ± 110.318	323.913 ± 96.229	0.231
IL8, median[IQR]		16.700 [11.500, 26.700]	15.400 [11.300, 26.300]	17.000 [12.300, 27.400]	0.461
Surgical approach, *n* (%)					
	indirect	85 (41.667)	44 (43.564)	41 (39.806)	0.586
	direct	119 (58.333)	57 (56.436)	62 (60.194)	

SD, standard deviation; BMI, body mass index; IQR, interquartile range; mRS, Modified Rankin Scale score; WBC, white blood cell; RBC, red blood cell; HGB, hemoglobin; PLT, platelet; PCT, procalcitonin; CHO, total cholesterol; TG, triglyceride; UA, Uric acid;Glu, glucose; Hcy, homocysteine; G1–G4, Level grading of IL-6 and TNF-α.

**Table 2 jcm-12-00823-t002:** Association of inflammatory markers IL-6 and TNF-α with postoperative modified Rankin Scale score ≥ 2.

Marker and Levels		Events [no., (%)]	Crude Model		Model 2		Model 2	
			OR (95% CI)	*p* Value	OR (95% CI)	*p* Value	OR (95% CI)	*p* Value
IL-6 (ng/L)	G1 (≤2.00)	42.029	Reference	–	Reference	–	Reference	–
	G2 (2.00–3.40)	49.275	1.34 [0.684, 2.623]	0.393	1.367 [0.664, 2.817]	0.396	1.321 [0.605, 2.886]	0.485
	G3 (3.40–4.83)	72.727	3.678 [1.491, 9.074]	0.005	4.522 [1.671, 12.234]	0.003	4.794 [1.669, 13.775]	0.004
	G4 (>4.83)	48.485	1.298 [0.564, 2.987]	0.539	0.994 [0.401, 2.463]	0.989	0.982 [0.371, 2.595]	0.97
P for trend	-	-	1.232 [0.948, 1.601]	0.119	1.179 [0.889, 1.564]	0.252	1.187 [0.880, 1.600]	0.261
TNF-α (ng/L)	G1 (≤6.24)	38.667	Reference	–	Reference	–	Reference	–
	G2 (6.24–8.10)	55.263	1.959 [1.025, 3.747]	0.042	2.017 [1.000, 4.067]	0.05	2.023 [0.957, 4.274]	0.065
	G3 (8.10–9.26)	55.556	1.983 [0.814, 4.829]	0.132	1.271 [0.490, 3.300]	0.622	1.2 [0.438, 3.288]	0.722
	G4 (>9.26)	65.385	2.996 [1.180, 7.610]	0.021	3.393 [1.211, 9.505]	0.02	4.616 [1.485, 14.355]	0.008
P for trend	-	-	1.432 [1.077, 1.903]	0.013	1.385 [1.021, 1.877]	0.036	1.467 [1.055, 2.040]	0.023

G1–G4, Level grading of IL-6 and TNF-α; OR, odds ratio.

## Data Availability

The raw data supporting the conclusions of this article will be made available by the authors, without undue reservation.

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
