# Peer review of "Circulating Inflammatory Cytokine Associated with Poor Prognosis in Moyamoya Disease: A Prospective Cohort Study"

_jcm, 2023, doi:10.3390/jcm12030823_

Round 1

Reviewer 1 Report

1. Methodology- The prognostication using only mRS is over-simplification as it emphasizes on the physical function mostly. This understates the importance of taking into account the frequency of TIA, headache and cognitive decline which are a major harbinger of poor quality of life in MMA patients. 

2. "Moyamoya syndromes include idiopathic MMD and secondary MMD"- Moyamoya Angiopathy includes Moyamoya Disease and Moyamoya syndrome or quasi Moyamoya. Moyamoya Syndrome has several associations other than inflammatory etiologies. Please look up the distinction and modifications should be made for the entire paragraph.

3. "This is due to the possibility that lower inflammatory factors can promote the development of MMD in individuals with the RNF213 mutation" needs explaining. The autoimmune hyperthyroidism model is well-accepted in Moyamoya Angiopathy. Kindly elaborate the explanation, it's not clear. Also, please ensure proper citations for this paragraph.

Reviewer 2 Report

Dear authors, Thank you for the privilege of reviewing your manuscript. I applaud the authors for taking the time to conduct this study, however there are several fundamental issues that currently prevent me from recommending this manuscript; I would recommend revision at this point. While I understand the following decision and feedback many be disappointing, my hope is the peer-review highlights weakness that can be strengthen; I believe with the appropriate changes the manuscript does hold promise. The following are the key issues precluding my ability to accept:

·      There are numerous variables that are associated with poor outcome that supersede that of inflammatory cytokines, however none of these variables were included in the multivariable analysis. Thus it is unclear the extent of bias in the results, with regards to the cytokines predicting prognosis independently. For the analysis I would recommend rather the authors conduct an analysis where they directly control (by stratifying groups) for the variables that are known to impact prognosis such as socioeconomic status, BMI, female sex, etc.

·      The authors should provide citations documenting variables that have been found to be risk factors for MMD, there are many variables that predict outcome which were omitted from this study—this should at least be mentioned in the discussion/limitations; the following citations discusses hypertension, diabetes, BMI, urban origin, female sex, and lower socioeconomic status as all being risk factors for MMD, that should be controlled for in any prognostication study.  

(A)  Sutton CXY, et al. Identification of associations and distinguishing moyamoya disease from ischemic strokes of other etiologies: A retrospective case-control study. Ann Med Surg (Lond). 2022 May 11;78:103771. doi: 10.1016/j.amsu.2022.103771. PMID: 35734698; PMCID: PMC9206914.

(B)  Socioeconomic and demographic disparities of moyamoya disease in the United States. Clin Neurol Neurosurg. 2020 May;192:105719. doi: 10.1016/j.clineuro.2020.105719. Epub 2020 Feb 4. PMID: 32045710.

Abstract:

-Please add space between “prognosis.TNF-α”

-Please add space between “(OR3.678,”

-Please add space between “CI[1.180,7.610]”

Introduction:

-Given the global importance of MMD, consider including a comparison to US and European incidence of MMD as MMD is relatively common in East Asian countries, while rarely discovered in the United States. Here is a suggested paper: Socioeconomic and demographic disparities of moyamoya disease in the United States. Clin Neurol Neurosurg. 2020;192:105719. doi:10.1016/j.clineuro.2020.105719

Clinical data collection:

-the risk factors is thorough and provides all vital information.

Statistical analysis:

-information provided is appropriate

Tables:

-Add space between “Table 1.Baseline C”

-In figure 1, reformat so (a) is properly shown in regards to TNF-alpha. Consider making the figure larger for better visualization of the graphs and formatting of the horizonal axis. 

-captions are appropriate

3.3:

-Please fix: p value formatting : p<0.001 and p<0.05

-Please add space between CI and ranges and OR and numerical values.

Round 2

Reviewer 1 Report

The authors have done a good work with the revision.